# Eating Habits and Sleep Quality during the COVID-19 Pandemic in Adult Population of Ecuador

**DOI:** 10.3390/ijerph18073606

**Published:** 2021-03-31

**Authors:** Patricio Ramos-Padilla, Verónica Dayana Villavicencio-Barriga, Haydeé Cárdenas-Quintana, Leonardo Abril-Merizalde, Angélica Solís-Manzano, Tannia Valeria Carpio-Arias

**Affiliations:** 1Research Group on Food and Human Nutrition, Escuela Superior Politécnica de Chimborazo, 060101 Riobamba, Ecuador; patricio.ramos@espoch.edu.ec (P.R.-P.); veronica.villavicencio@espoch.edu.ec (V.D.V.-B.); 2Gastronomy Career, Faculty of Public Health, Escuela Superior Politécnica de Chimborazo, 060103 Riobamba, Ecuador; 3Department of Nutrition, Universidad Nacional Agraria La Molina, 15012 Lima, Peru; hcardenasq@lamolina.edu.pe; 4Nutrition and Dietetics Career, Faculty of Public Health, Escuela Superior Politécnica de Chimborazo, 060101 Riobamba, Ecuador; dennys.abril@espoch.edu.ec; 5Research Group in Nutrition, Dietetics, Biotechnology and Food Analysis, Universidad Estatal de Milagro, 091701 Milagro, Ecuador; asolism2@unemi.edu.ec

**Keywords:** eating habits, sleep quality, COVID-19, diet

## Abstract

Confinement due to COVID-19 has brought important changes in people’s lives as well as in their eating and resting habits. In this study we aimed at exploring the eating habits and sleep quality of the adult population of Ecuador during the mandatory confinement due to the COVID-19 pandemic. This is a cross-sectional study, which used an online survey that included questions about eating habits and sleeping habits in adults (*n* = 9522) between 18–69 years old. The Pittsburg sleep quality questionnaire validated for the Hispanic population was used, and questions about dietary habits. The statistical test Chi-square statistical test was used to analyze the data. The results show that sleep quality differs according to sex, being worse in women, both in all components of sleep quality and in the total score (*p* < 0.001). Women had greater changes in the habitual consumption of food compared to men (24.24% vs. 22.53%), and people between 18 and 40 years of age decreased their food consumption in relation to people >40 years (24.06% vs. 17.73%). Our results indicate that mandatory confinement due to the COVID-19 pandemic in Ecuador has generated changes in the eating habits and sleep quality in the adult population sampled, and these changes are more noticeable in women and young adults. These changes offer an important alert for the health system and further, advice for the implementation of future public health policies.

## 1. Introduction

The new coronavirus (Sars-CoV2) emerged sparking a pandemic of acute respiratory syndrome (COVID-19) in humans that has caused, as of 1 December 2020, more than 63,890,000 people to be affected, with over 1,480,000 deaths worldwide across five continents [1,2,3,4], and it is the main contributing factor to the international public health emergency [5,6]. In Ecuador, according to the National Service for Risk and Emergency Management (NSREM), as of 12 March 2021, there have been 299,216 confirmed cases and 16,193 possible deaths [7].

The advance of the highly contagious coronavirus, the increase in the number of deaths, the overload of the health services and the absence of etiological treatment, forced government authorities to declare a state of health alarm and order the confinement of citizens [8]. Beyond stresses inherent in the illness itself, mass home confinement directives are new for the general population and raise concerns about how people will react individually and collectively [9,10,11,12], because the confinement periods imply daily routine and lifestyle changes for the population [13]. This social isolation, together with the economic difficulties associated with job losses, in addition to reduction in labor availability for the food sector, have put the food security and nutritional status of the population at risk [14,15,16]. Previous studies have been published on the food basket and its differential evolution during the period of confinement due to COVID-19. For example, limited access to daily grocery shopping may lead to a reduction in the consumption of fresh foods, especially fruit, vegetables and fish, in favor of highly processed ones, such as convenience foods, junk foods, snacks, and ready-to-eat cereals, which tend to be high in fats, sugars, and salt [17,18]. In the same way, they have drawn attention to the potential risks of this situation in relation to health, emotional balance, coexistence [19,20], the alteration of eating habits, and an increase in sedentary lifestyle [21,22]. Several studies identify as main metabolic consequences, increases in insulin resistance, total body fat, abdominal fat and inflammatory cytokines. All these factors have been strongly associated with the development of metabolic syndrome, which in turn increases the risk of multiple chronic diseases [23]. Likewise, it is relevant to remember that the selection, purchase, storage and handling of food can favor the spread of COVID-19 [24,25].

Additionally, it should be mentioned that, although no food or medicine can reduce or inactivate the viral load [26]; however, the consumption of a healthy diet helps to strengthen and maintain an immune system in balance with health [27,28] that contributes to dealing with infectious diseases such as COVID-19 [29].

In another way, sleep habits in the population have also changed during the COVID-19 pandemic [29]. The sleep–wake cycle is subject to multiple factors, mainly exposure to daylight and darkness at night, the latter increasing the levels of melatonin, a hormone that plays a key role in the regulation and initiation of sleep, that can be affected by habits such as sleeping with a mobile phone [30,31]. Other factors involved are the same mealtimes and daytime physical activity, the latter occurring both at low levels (depression or forced confinement), and at high levels (stress, work overload or intense night-time exercise); consequently, sleep patterns are negatively affected in the context of mandatory confinement [29]. In this sense, stress implies greater psychological and physiological activation in response to daily demands. Increased hypothalamic-pituitary-adrenal (HPA) axis function is known to be associated with shortened and fragmented sleep, and a possible reduction in N3 sleep phase (deepest phase of sleep) [30]. In this context, shortened sleep or poor sleep quality have been associated with unfavorable health effects, such as mental health problems and overweight [32,33,34], with some studies reported in the Ecuadorian population [35].

We set out to describe the eating habits and quality of sleep of the adult population of Ecuador during the mandatory confinement due to the COVID-19 pandemic.

## 2. Materials and Methods

### 2.1. Study Type and Setting

This was a non-experimental cross-sectional study. With regard to the setting of the study, 9522 Ecuadorian individuals, men and women, aged between 18 and 69 years participated.

### 2.2. Instruments and Variables

An online survey was created, which was reviewed by 4 nutrition experts and then applied to 30 Ecuadorian adults as a pilot test. All comments, such as the wording and the order of the questions, were considered to create an improved version of the survey (data not shown). The survey was self-administered only once during June and July of 2020 and the data were collected using Google Forms, (Google, LLC Mountain View, CA, USA). Participant responses are anonymous and confidential in accordance with Google’s privacy policy [36].

The survey link was disseminated through the official media of the Ecuadorian Universities: Universidad Estatal de Milagro (UNEMI) and the Escuela Superior Politécnica de Chimborazo (ESPOCH). In addition, the survey was disseminated on social networks such as Facebook, Instagram and through WhatsApp (Facebook, Inc. Merlo Park, CA, USA). The online survey had 4 sections: (1) Presentation, aim and informed consent form, (2) General and sociodemographic characteristics of the population, (3) Questions about eating habits (framed in the context of the current situation of the COVID-19 pandemic), (4) A validated questionnaire for the Hispanic population called the Pittsburgh Sleep Quality Index (PSQI) [37], which was used to measure the quality and patterns of sleep in the last month.

In order to assess the reliability of the questionnaire that was used in this study, the Cronbach’s alpha was calculated, which was 0.77. The PSQI is the gold standard questionnaire for evaluating subjective sleep quality and it is made up of 21 items divided into seven factors. The scores for each factor are added, obtaining a score that ranges from 0 to 21 points. For this study, a dichotomization of this variable was performed using the median value of the population score (4.0 points) as the cut-off point.

### 2.3. Statistical Analysis

The data were analyzed using the statistical program R (R Foundation for Statistical Computing, Vienna, Austria) and RStudio (RStudio PBC Boston, MA, USA). For the analysis of quantitative variables, calculations of measures of central tendency, and of dispersion, were carried out. For variables of ordinal or nominal qualitative scale, frequencies and percentages were calculated. To establish relations between variables, Chi-square statistical test with Bonferroni adjustment was used for components of sleep quality according to gender. To establish relationships between changes in habitual food intake according to demographics, Chi-square statistical was used for characteristics. To establish statistically significant differences, a *p*-value < 0.05 was considered.

### 2.4. Ethical Considerations

Prior to completing the survey, the participants read and provided their informed consent at the beginning of the online form. It was requested that the survey be answered only by people of legal age (over 18 years old), so the records (*n* = 60) of minors were eliminated from the database. None of the personal data is identifiable.

This study was developed following the Declaration of Helsinki, regarding work with human beings and according to the “Singapore Declaration on Research Integrity”.

## 3. Results

The data of a total of 9522 adults between 18 and 69 years old was analyzed. In the sample, 69.4% of participants belong to the female sex and 71.9% reside in urban areas. Regarding occupation, 85.7% were from the group “housewives-students-retirees” and 73.4% were single. The average number of inhabitants in the household was 4.7 ± 1.8 members (Table 1).

Regarding the way of acquiring food during mandatory confinement due to the pandemic, the majority of the population (56.0%) did so in the local market or store, while only 3.8% did so through delivery services to the home (Table 2).

Regarding habits in the consumption of food, nutritional supplements and beverages; 50.4% of participants indicated that they had changed their usual mealtimes, 16.0% stopped consuming some food because they considered it harmful, while 44.0%, 41.4% and 31.6%, respectively, increased the consumption of some foods, supplements or beverages, as they considered it to be beneficial (Table 3).

In relation to changes in habitual food consumption, women presented a more pronounced decrease in consumption than men (24.24% vs. 22.53%). The rural population showed a more pronounced decrease in consumption in relation to the urban population (28.05% vs. 22.02%). The population 18–40 years old presented a greater decrease in consumption compared to the population > 40 years old (24.06% vs. 17.73%). The housewives-students-retirees population showed a greater decrease in consumption in relation to other occupations (24.67% vs. 21.83% for teachers-researchers vs. 16.28% for employees-entrepreneur-business owner). These differences are statistically significant within groups (*p* < 0.05) (Table 4).

Differences were found by sex in most of the components of sleep quality described below (Table 5): Subjective sleep quality is pretty bad in women, 15.2% versus 12.6% in men (*p* < 0.001). Sleep duration: 17.4% of women sleep less than 5 h versus men (19.7%) (*p* < 0.001). Sleep efficiency, measured according to the Pittsburgh questionnaire protocol, was measured in percentages, where a value less than 65% is considered the lowest sleep efficiency. In this sense, women had a lower sleep efficiency of 3.6%, compared to men (3.0%). These differences were not statistically significant (*p* = 0.265).

Women present sleep disturbances less than once a week in 69.2% vs. men who present them less frequently in 62.9% (*p* < 0.001). The comparison of the use of hypnotic medication less than once a week was statistically significant between women (less use) and men (*p* = 0.011), as 4.9% of women use said medication in relation to men (6.2%). Women have diurnal dysfunction less than once a week in 40.4% vs. men (34.5%) (*p* < 0.001). Results of sleep latency are not presented because it does not have a practical interpretation.

Finally, the quality of sleep of the population, during compulsory confinement differs according to sex. It is of poor quality for 51% of women compared to 47% of men, this difference was statistically significant (*p* < 0.05) (Table 5).

## 4. Discussion

### 4.1. Changes in Shopping Habits and Type of Food Consumed

During data collection (June–July 2020) most people bought their food in the local market or store, despite the mobility restrictions established by the NSREM, so it could be assumed that the confinement for COVID-19 did not represent a significant risk to people’s food safety. A recent study in Spain also ensures that the diet of this population during the confinement has been framed within food security [38]. Our findings would not be in accordance with similar studies carried out in Colombia [39], as well as in some European, Asian and African countries [40] where diet habits during confinement have been less healthy. The Food and Agriculture Organization of the United Nations (FAO) and the Economic Commission for Latin America and the Caribbean (ECLAC) have warned about the risk that the world’s population takes from having less nutritious and non-fresh food intakes during confinement due to COVID-19 [41].

Regarding the change in the type of food consumed, the data indicates notable variation in the consumption of food, supplements and beverages, as well as the abandonment of some foods; however, these results could not support the benefit in the COVID-19 treatment [40,41,42].

On the other hand, the result of the changes in the population’s food consumption reflects that the most vulnerable groups who decrease food consumption are: women, inhabitants of rural areas, young adults and students, housewives or retirees. These results would be useful to identify vulnerable groups in food insecurity of the Ecuadorian population.

### 4.2. Repercussions of Changes in Feeding Times on People’s Health during the COVID-19 Pandemic

Around half of the surveyed population changed their usual mealtimes. This fact would affect the nutritional status of individuals as reported by Kalheova [43]; skipping breakfast and increasing the consumption of lunches and dinners cause a significant increase in body mass index (*p* < 0.001). Likewise, other investigations [44,45] have shown that changing mealtimes affects body composition, and it can lead to the development of cardio-metabolic diseases and diabetes. Other research reported that individuals who do not eat breakfast, and consume snacks before dinner and before bed, are associated with type 2 diabetes mellitus [46]. Consequently, it is important to highlight that these pathologies play an important role in the infection and prognosis due to COVID-19 [46].

It is important to highlight other studies that show that an unusual feeding time can produce alteration of the circadian system that could have harmful consequences for human health [47]. The energy distribution of the diet would be an important predictor for losing or gaining weight, where people who eat high-energy meals for lunch or dinner, gain weight and may become overweight and obese [48]. According to the results found, the change in mealtimes that people have suffered during the pandemic could be considered as a predisposing risk factor for weight gain, and we believe the findings obtained point to potential future research in Ecuador.

### 4.3. Impact on Changes in Sleep Schedules during the COVID-19 Pandemic

Sleep habits can be measured both through aim methods [49], as well as through psychometric questionnaires that analyze aspects such as sleep duration in isolation, which can be imprecise [50]. Sleep Quality includes in its definition quantitative aspects of sleep, such as duration, latency, number of nocturnal awakenings, and qualitative aspects that make it a more accurate measurement despite its subjective evaluation method [51].

In this sense, the health emergency has conditioned the population to new life habits and among them, sleep habits [38]. This is one of the physiological processes that is related to the adequate metabolic state, an indicator that when altered, contributes to premature aging, cancer, and shortened life expectancy [52].

The change in sleep habits can be strongly related to the timing of meals, becoming a risk factor for the gain and progression of body weight and comorbidities [53]. In addition, it should be considered that several physiological processes respond to a circadian clock in a 24-h cycle, as well as several genes that participate in metabolic processes which are also circadian [54].

Previous studies conducted in Ecuador have shown important associations between sleep disorders and strokes [35] as well as metabolic syndrome [55]. However, longitudinal studies must be carried out to confirm the cause–effect relationships of these events.

Eating habits and sleep quality were evaluated in this study subjectively. For this reason, it is recommended for future studies to implement objective methods that allow for determining possible, more exact, causal relationships.

In this study, a higher prevalence of problems related to sleep quality (total score and most of the components assessed within sleep quality) was found in women, compared to men. In this sense, it can be said that stay at home directive affects this population group differently, so health policies and interventions must consider the situation of this particular group.

### 4.4. Study Limitations

This study used an online survey as a data collection instrument, so the population with low economic resources, or older adults, may not have participated in this study due to internet access problems; however, the authors of this work would like to highlight the sample size of this study as a positive counterpoint. In addition, the diet analysis was qualitative, so it is recommended to carry out future studies with a quantitative approach.

## 5. Conclusions

The mandatory confinement due to the COVID-19 pandemic in Ecuador has induced changes in eating habits and in the quality of sleep of the adult population. Of every 100 people, 56 obtained their food in the local market or store, which implied leaving home. More than half of the population changed their usual mealtimes, 16.0% stopped consuming some food because they considered it harmful, and 44.0%, 41.4% and 31.6%, respectively, increased the consumption of some food, supplement or beverage, considering them beneficial. Women showed a greater decrease in food consumption than men and, the rural population had a greater decrease in consumption than the urban population. The population between 18 and 40 years old showed greater decrease in consumption than the population > 40 years, the population of the group housewives-students-retirees showed a greater decrease in consumption in relation to other occupations. In relation to the quality of sleep of the population, it was poor in 51% of the women vs. 47% of the men, showing statistically significant differences in most of the components of sleep quality.

Although this study was carried out in the Ecuadorian population, the findings could represent important topics for the health system in general, encouraging more studies in local contexts and decision making specifically tailored to promoting women’s health. 

In this new stage of gradual return to normality, it is necessary to reinforce positive changes, such as a healthier diet and encouraging the purchase of food from locations, as much as possible, in so-called neighborhood stores and local markets that comply with all the biosecurity standards. In addition, the progressive return to the practice of physical activity in its different forms, and all other actions that favor emotional balance should be strongly motivated.

## Figures and Tables

**Table 1 ijerph-18-03606-t001:** Demographic characteristics studied. Adult population, both sexes.

Characteristics				Frequency	Percentage
Sex	Woman			6610	69.4
	Man			2912	30.6
Residence area	Rural			2674	28.1
	Urban			6848	71.9
Occupation	Housewife-student-retiree			8156	85.7
	Teacher-researcher			426	4.5
	Employee-entrepreneur-business owner			940	9.9
Marital status	Married-common law marriage			2227	23.4
	Separated-divorced-widowed			303	3.2
	Single			6992	73.4
	**Minimum**	**Maximum**	**Mean**	**SD**	**Asymmetry**
Age	18	69	24.91	7.647	1.802
Inhabitants at home	1	12	4.71	1.849	1.049

SD = Standard Deviation.

**Table 2 ijerph-18-03606-t002:** Acquisition of food during mandatory confinement because of the COVID-19 pandemic.

		Frequency	Percentage
Way acquisition of food	Home delivery	358	3.8
	Local market or store	5334	56.0
	Supermarkets	1418	14.9
	All of the above	2412	25.3
	Total	9522	100.0

**Table 3 ijerph-18-03606-t003:** Eating habits, nutritional supplements and beverages intake; total population.

N = 9522		Frequency	Percentage
Changed usual mealtimes	No	4722	49.6
	Yes	4800	50.4
Stopped consumption of any food because it was considered harmful	No	7994	84.0
	Yes	1528	16.0
Increased consumption of any food because it was considered beneficial	No	5330	56.0
	Yes	4192	44.0
Increased consumption of any supplement because it was considered beneficial	No	5582	58.6
	Yes	3940	41.4
Increased consumption of any beverage because it was considered beneficial *	No	6511	68.4
	Yes	3011	31.6

* Except alcoholic beverages.

**Table 4 ijerph-18-03606-t004:** Changes in habitual food intake; total population, and by demographic characteristics.

		Increased	Decreased	Maintained	*p*-Value
Total	Frequency	1980	2258	5284	
	Percentage	20.79	23.71	55.49	
Woman	Frequency	1455	1602	3553	<0.0001 *
	Percentage	22.01	24.24	53.75	
Man	Frequency	525	656	1731	
	Percentage	18.03	22.53	59.44	
Rural	Frequency	469	750	1455	<0.0001 *
	Percentage	17.54	28.05	54.41	
Urban	Frequency	1511	1508	3829	
	Percentage	22.06	22.02	55.91	
Housewife-student-retiree	Frequency	1630	2012	4514	<0.0001 *
	Percentage	19.99	24.67	55.35	
Teacher-researcher	Frequency	105	93	228	
	Percentage	24.65	21.83	53.52	
Employee-entrepreneur-business owner	Frequency	245	153	542	
	Percentage	26.06	16.28	57.66	
Married-common law marriage	Frequency	486	536	1205	0.4528
	Percentage	21.82	24.07	54.11	
Separated-divorced-widowed	Frequency	68	66	169	
	Percentage	22.44	21.78	55.78	
Single	Frequency	1426	1656	3910	
	Percentage	20.39	23.68	55.92	
18–40 years	Frequency	1865	2166	4972	0.0043 *
	Percentage	20.72	24.06	55.23	
>40 years	Frequency	115	92	312	
	Percentage	22.16	17.73	60.12	

* Statistically significant.

**Table 5 ijerph-18-03606-t005:** Sleep quality in the study population by sex.

	Woman		Man		*p*-Value
	*n* = 6610		*n* = 2912		
	N	%	N	%	
Subjective sleep quality					<0.001 *
Very good	2036	30.8	959	32.9	
Pretty good	3131	47.4	1473	50.6	
Pretty bad	1008	15.2	368	12.6	
Very bad	435	6.6	112	3.8	
Sleep duration					<0.001 *
>7 h	2966	44.9	1182	40.6	
6–7 h	1338	20.2	657	22.6	
5–6 h	1156	17.5 *^a^	498	17.1 *^a^	
<5 h	1050	17.4	575	19.7	
Sleep efficiency					0.265
>85%	5050	76.4 *^a^	2221	76.3 *^a^	
75–84%	0	0 *^a^	0	0 *^a^	
65–74%	1323	20.0 *^a^	604	20.7 *^a^	
<65%	237	3.6 *^a^	87	3.0 *^a^	
Sleep disturbance					<0.001 *
No time in the month	2033	30.8	1079	37.1	
Less than once a week	4577	69.2	1833	62.9	
1–2 times a week	0	0	0	0	
3 or more times a week	0	0	0	0	
Use of hypnotic medication					0.011 *
No time in the month	5988	90.6 *^a^	2629	90.3 *^a^	
Less than once a week	323	4.9	180	6.2	
1–2 times a week	171	2.6 *^a^	60	2.1 *^a^	
3 or more times a week	128	1.9 *^a^	43	1.5 *^a^	
Daytime dysfunction					<0.001 *
No time in the month	3214	48.6	1659	57	
Less than once a week	2672	40.4	1005	34.5	
1–2 times a week	612	9.3	215	7.4	
3 or more times a week	112	1.7	33	1.1	
Sleep quality (Total score)					<0.001 *
Poor quality of sleep	3372	51	1369	47	
Good sleep quality	3238	49	1543	53	

Sleep quality measured according to the Pittsburgh Sleep Quality Index (PSQI), the six factors of the PSQI are presented: subjective sleep quality, sleep duration, sleep efficiency, sleep disturbances, use of hypnotic medication and daytime dysfunction. Sleep latency does not occur because it does not have a practical interpretation. * The differences between all groups are statistically significant. *^a^ The differences between the groups indicated with the subscript are not statistically significant.

## Data Availability

The database of this study can be made available upon request by emailing the corresponding author.

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
