# Peer review of "Eating Habits and Sleep Quality during the COVID-19 Pandemic in Adult Population of Ecuador"

_ijerph, 2021, doi:10.3390/ijerph18073606_

Round 1
Reviewer 1 Report
Authors have conducted research on the eating habits and sleep quality during COVID-19 pandemic in Adult population of Ecuador. This is a very interesting topic and is very relevant considering our current situation regarding COVID-19. However, I have suggestions to improve this manuscript.
Some suggestions for improvement:
Abstract:
Line 21: There needs to be a dash inserted between cross and sectional. “cross-sectional”
Line 24: Please consider rewording to “ Adults (n=9522) between the ages of 18 and 69 years were recruited to participate in the study”.
When you say the participants were analysed it becomes confusing for the reader.
Line 25: Chi-square should be written in full. Maybe to improve the way you say this: Appropriate statistical tests (Chi-sqaure ……) were used to analyse the data.
Line 31: There needs to be a space between the word population and > sign.
Line 32: The conclusion is very vague. In way has it changed. Adjust the sentence to make it more specific.
Introduction
Lines 37-40: Please summarise to one sentence or try and shorten
e.g. The coronavirus-2019 also known as COVID-19 is a global pandemic and is the main contributor to the international public health emergency.
Lines 43-46: “The advance…..citizens” This sentence does not read well. Please consider splitting the sentence into two sentences so that the meaning is not lost.
Line 48: Authors mention that there have been other studies published on foods baskets and the confinement period. Give more details on what was found. Did these studies find a decrease in healthy food, increase in fast foods, poor sleeping patterns, high rates of unemployment, what were the risks to health found? Was the health risks specific to the COVID-19 virus or diseases of lifestyle such as obesity, DM, HPT etc.
Specifically, in Ecudador have there been any studies conducted on what foods are commonly consumed?
Doing evening work tend to snack more etc. Also link sleep patterns to eating.
Sleep also effects eating patterns. People who are stressed eat more. If individuals consume foods later at night, it interferes with the release of melatonin. Melatonin is needed to have a healthy sleep pattern.
When one is under stress cortisol is released , this can disrupt seep patterns and lead to eating more food etc.
A good basis for the introduction, please consider strengthening the introduction.
Methodology
Line 81-84: You mention you built a survey, does this mean you have a programme that you designed for the survey to run? Please split this sentence as there are too many ideas in one sentence. Did this online survey make use of standardized questions?
How did you select the pilot study participants? Why did you select 30? How did you ensure that the pilot study participants did not participate in the main study.
Was the actual survey sent out or a link to the survey?
Line 92: Valited? Is this correct
Line 104: Should be chi-square
Ethical consideration
With your online survey, if the participant did not give consent what happened? Where they redirected to a page that said we have noted you have not given consent and do not want to participate, thank you for your time. Was there some sort of structure in place that prevented them from obtaining the survey questions?
Did you receive any gatekeeper’s permission to conduct the study?
Results
Line 139: Add a space between population and > sign.
Tables:
Please check consistency of the tables. Some parts are in bold and CAPS and others not. Ensure all tables are consistent.
Table 3: what is SI?
Table 5: Please check your key matches the table * under P value
Discussion
Line 181: add the word “of” before COVID-19
Line 191: Remove the full stop from the heading
Line 193: We don’t use her, he or us in scientific writing. Rather say the author.
Well done for linking the eating patterns to diseases of lifestyle/noncommunicable disease (DM, obesity etc)
Well done with including the study limitations and implications for research and practice. How will one reinforce positive changes? Education?
Conclusion
Very brief. What is the way forward? What are the main changes that it has induced?
Author Response
Authors have conducted research on the eating habits and sleep quality during COVID-19 pandemic in Adult population of Ecuador. This is a very interesting topic and is very relevant considering our current situation regarding COVID-19. However, I have suggestions to improve this manuscript.
Some suggestions for improvement:
THANKS FOR YOUR SUGGESTIONS
Abstract:
Line 21: There needs to be a dash inserted between cross and sectional. “cross-sectional”
The change was made.
Line 24: Please consider rewording to “Adults (n=9522) between the ages of 18 and 69 years were recruited to participate in the study”.
When you say the participants were analysed it becomes confusing for the reader.
The change was made.
Line 25: Chi-square should be written in full. Maybe to improve the way you say this: Appropriate statistical tests (Chi-square ……) were used to analyse the data.
The change was made.
Line 31: There needs to be a space between the word population and > sign.
The change was made.
Line 32: The conclusion is very vague. In way has it changed. Adjust the sentence to make it more specific.
The conclusion was expanded.
Introduction
Lines 37-40: Please summarise to one sentence or try and shorten
e.g. The coronavirus-2019 also known as COVID-19 is a global pandemic and is the main contributor to the international public health emergency.
Has been summarized.
Lines 43-46: “The advance…..citizens” This sentence does not read well. Please consider splitting the sentence into two sentences so that the meaning is not lost.
The change was made.
Line 48: Authors mention that there have been other studies published on foods baskets and the confinement period. Give more details on what was found. Did these studies find a decrease in healthy food, increase in fast foods, poor sleeping patterns, high rates of unemployment, what were the risks to health found? Was the health risks specific to the COVID-19 virus or diseases of lifestyle such as obesity, DM, HPT etc.
Specifically, in Ecudador have there been any studies conducted on what foods are commonly consumed?
Doing evening work tend to snack more etc. Also link sleep patterns to eating.
Sleep also effects eating patterns. People who are stressed eat more. If individuals consume foods later at night, it interferes with the release of melatonin. Melatonin is needed to have a healthy sleep pattern.
When one is under stress cortisol is released , this can disrupt seep patterns and lead to eating more food etc.
A good basis for the introduction, please consider strengthening the introduction.
The introduction has been improved based on the suggestions.
Methodology
Line 81-84: You mention you built a survey, does this mean you have a programme that you designed for the survey to run? Please split this sentence as there are too many ideas in one sentence.
It is specified that an online survey was created through Google Forms, which is survey management software that is included as part of the free web-based Google document editors package offered by Google.
Did this online survey make use of standardized questions?
It is specified that questions on eating habits were developed, framed in the current situation of the COVID-19 pandemic, and a validated questionnaire for Hispanic population called the Pittsburgh Sleep Quality Index (PSQI) was used. To assess the reliability of the questionnaire used in this study, Cronbach's alpha was calculated, which was 0.77.
How did you select the pilot study participants? Why did you select 30? How did you ensure that the pilot study participants did not participate in the main study?
It was socialized in social networks inviting to participate in the pilot study, it was possible to recruit 30 volunteers. Participants in the pilot study were not controlled to participate in the main study.
Was the actual survey sent out or a link to the survey?
It is specified that the survey link was disseminated.
Line 92: Valited? Is this correct
It was changed to validated.
Line 104: Should be chi-square
It was changed by chi-square.
Ethical consideration
With your online survey, if the participant did not give consent what happened? Where they redirected to a page that said we have noted you have not given consent and do not want to participate, thank you for your time. Was there some sort of structure in place that prevented them from obtaining the survey questions?
If the participant did not consent, they were redirected to the final page that said Click Submit to Finish. If you wish, you can verify it by clicking on the following link:
https://docs.google.com/forms/d/e/1FAIpQLSfq7BMErQsWnraZsVmQZzCytWwwh_x965KZ2YoHaxWhkL2Tfw/viewform?usp=sf_link
Did you receive any gatekeeper’s permission to conduct the study?
It is specified that the responses of the participants are anonymous and confidential in accordance with Google's privacy policy.
Results
Line 139: Add a space between population and > sign.
The change was made.
Tables:
Please check consistency of the tables. Some parts are in bold and CAPS and others not. Ensure all tables are consistent.
Formatting of all tables was corrected.
Table 3: what is SI?
Sorry, it was a problem with the translation, it has been corrected to Yes.
Table 5: Please check your key matches the table * under P value
Table was corrected. The key was verified to match the table and the table footer.
Discussion
Line 181: add the word “of” before COVID-19
The change was made.
Line 191: Remove the full stop from the heading
The change was made.
Line 193: We don’t use her, he or us in scientific writing. Rather say the author.
The change was made.
Well done for linking the eating patterns to diseases of lifestyle/noncommunicable disease (DM, obesity etc)
Well done with including the study limitations and implications for research and practice. How will one reinforce positive changes? Education?
Conclusion
Very brief. What is the way forward? What are the main changes that it has induced?
The conclusion was expanded based on the suggestions.
Reviewer 2 Report
After analysing the manuscript "Eating Habits and Sleep Quality During the COVID-19 Pandemic in Adult Population of Ecuador" I can indicate that the study presented is interesting and relevant to the scientific community. Among the strengths of the study is the sample studied. It is very large. Even so, there are several issues that the authors must address in order for it to be considered for publication. I list the issues:
1.- The theoretical framework is very scarce. It has only 16 references. They should expand the theoretical framework (introduction) with a total of at least 35 references. I leave you a manuscript that can serve as a reference to expand the present study. https://www.mdpi.com/1660-4601/17/10/3697
2.- More explanation of the instrument is needed. In order for the results to be considered valid, it is necessary to prove that the instrument is valid and reliable. I therefore ask the authors to specify the tests and the values achieved to ensure that the instrument is valid and reliable.
3.- I recommend the authors to establish future lines of research.
I would like to congratulate the authors for the work presented. I hope that they will heed all the recommendations set out.
Author Response
After analysing the manuscript "Eating Habits and Sleep Quality During the COVID-19 Pandemic in Adult Population of Ecuador" I can indicate that the study presented is interesting and relevant to the scientific community. Among the strengths of the study is the sample studied. It is very large. Even so, there are several issues that the authors must address in order for it to be considered for publication. I list the issues:
We welcome the reviewer's suggestions and comments, which allow us to improve our work.
We have also added current and relevant references in the introduction section.
The answers to each of the suggestions are presented below.
1.- The theoretical framework is very scarce. It has only 16 references. They should expand the theoretical framework (introduction) with a total of at least 35 references. I leave you a manuscript that can serve as a reference to expand the present study. https://www.mdpi.com/1660-4601/17/10/3697
Thanks for your observation, we have added the suggested information
2.- More explanation of the instrument is needed. In order for the results to be considered valid, it is necessary to prove that the instrument is valid and reliable. I therefore ask the authors to specify the tests and the values achieved to ensure that the instrument is valid and reliable.
We have added the information about de values achieves to ensure that the instrument is valid.
3.- I recommend the authors to establish future lines of research.
Thanks for your observation, we have added the suggested information
I would like to congratulate the authors for the work presented. I hope that they will heed all the recommendations set out.
We appreciate the comments of the Reviewer, we have taken into account each of the recommendations and we are sure that they will allow us to improve the quality of our work.
Reviewer 3 Report
The article titled “Eating Habits and Sleep Quality During the COVID-19 Pan-2 demic in Adult Population of Ecuador” is well constructed and reads well. Seems like it's quite relevant to the current scenario and in general related to effects on sleep including stress and isolation. The paper's synthesis seemed all right to me. Overall, the results are well presented and discussed. Although, there still persists some issues/suggestions that need to be addressed before consideration for publication.
- Lines 37-40, the first paragraph of introduction is quite complex and not making much sense, please try to simplify and improve this.
- Line 72, stress. insomnia is not making much sense either.
- Line 92, should read validated rather valited.
- In tables, the major categories are not presented uniformly. Please try to present in uniformity.
- Table 3 denotes Si which is not explained well.
- Reference 35, its has a missing volume and the page number. make sure to feed it if available?
- Grammatically, few sentences are written in a third-person e,g., line 232-234. Try to carefully check and revise the manuscript in this aspect.
Author Response
The article titled “Eating Habits and Sleep Quality During the COVID-19 Pan-2 demic in Adult Population of Ecuador” is well constructed and reads well. Seems like it's quite relevant to the current scenario and in general related to effects on sleep including stress and isolation. The paper's synthesis seemed all right to me. Overall, the results are well presented and discussed. Although, there still persists some issues/suggestions that need to be addressed before consideration for publication.
We appreciate the comments of the Reviewer, we consider the issue relevant, both for being an important social problem and for being a subject of analysis for the scientific community. We have added current and relevant references in the introduction section. We have also improved the conclusions of the work
The answers to each of the suggestions are presented below.
- Lines 37-40, the first paragraph of introduction is quite complex and not making much sense, please try to simplify and improve this.
Thanks for the correction, we have improved the information and writing
- Line 72, stress. insomnia is not making much sense either.
Thanks for the correction, we have made the change
- Line 92, should read validated rather valited.
Thanks for the correction, we have made the change
- In tables, the major categories are not presented uniformly. Please try to present in uniformity.
Thanks for the correction, we have made the change
- Table 3 denotes Si which is not explained well.
Thanks for the correction, we have made the changes
- Reference 35, its has a missing volume and the page number. make sure to feed it if available?
We have added the information to reference number 35
- Grammatically, few sentences are written in a third-person e,g., line 232-234. Try to carefully check and revise the manuscript in this aspect.
Thanks for the correction, we have made the change
Reviewer 4 Report
The topic addressed in the study carried out by Ramos-Padilla et al is very emergent and relevent. However, several issues need to be taking into consideration before its consideration for publication in IJERPH.
The English language of all the manuscript needs extensive editing. Here are just some exemples of inappropriate use of Scientific English:
“Methods: Cross sectional study, an online survey was developed where eating habits and sleeping habits were explored during confinement to the COVID-19 pandemic”
“Non-experimental cross-sectional study. Regarding the setting of the study, 9,522 Ec- 78 uadorian individuals, men and women, aged between 18 and 69 years participated.”
The background mentioned in the abstract is not appropriate. The authors indicate the goals of the study and not the background.
“Prior to completing the survey, the participants read and pro- 23 vided their informed consent” – This information is not needed to provide in the abstract.
The conclusions pointed out by the authors in the abstract as well as in the section 5 don’t bring anything new to the previous knowledge. How does the mandatory confinement in Ecuador due to COVID-19 has induced changes in eating habits and sleep quality in the adult population? What changes? In each way?
A more in-depth literature review is needed in the Introduction regarding the situation in Ecuador during this pandemic period as well as their eating habits and sleep quality before the pandemic.
The authors need to clearly point out why this study is relevant and is useful for the international community.
It would be useful if the authors could provide the online survey they built.
The discussion needs to be extended, mainly regarding the Impact on changes in sleep schedules during the COVID-19 pandemic. More studies should be provided and discussed.
Author Response
REVISOR 4
The topic addressed in the study carried out by Ramos-Padilla et al is very emergent and relevant. However, several issues need to be taking into consideration before its consideration for publication in IJERPH.
We greatly appreciate the comments of the reviewers which allow our work to improve. We have added current and relevant references in the introduction section. We have also improved the conclusions of the work, as well as we have provided deeper explanations of the methodological section and a better writing of the results.
The answers to each of the suggestions are presented below.
The English language of all the manuscript needs extensive editing. Here are just some examples of inappropriate use of Scientific English:
We have reviewed all the grammatical and linguistic writing of the document
“Methods: Cross sectional study, an online survey was developed where eating habits and sleeping habits were explored during confinement to the COVID-19 pandemic”
“Non-experimental cross-sectional study. Regarding the setting of the study, 9,522 Ec- 78 uadorian individuals, men and women, aged between 18 and 69 years participated.”
The background mentioned in the abstract is not appropriate. The authors indicate the goals of the study and not the background.
We have increased background section in the summary
“Prior to completing the survey, the participants read and pro- 23 vided their informed consent” – This information is not needed to provide in the abstract.
Thank you for your recommendation, we have removed this sentence from the summary
The conclusions pointed out by the authors in the abstract as well as in the section 5 don’t bring anything new to the previous knowledge. How does the mandatory confinement in Ecuador due to COVID-19 has induced changes in eating habits and sleep quality in the adult population? What changes? In each way?
Thanks for the recommendation, we have made changes to the conclusions of the document
A more in-depth literature review is needed in the Introduction regarding the situation in Ecuador during this pandemic period as well as their eating habits and sleep quality before the pandemic.
Information has been increased in the introduction and discussion about the quality of sleep and eating habits before the pandemic
The authors need to clearly point out why this study is relevant and is useful for the international community.
We have made the suggested changes in the "conclusions" section
It would be useful if the authors could provide the online survey they built.
The survey link is presented below, the survey is in Spanish. For your review we have opened it again, however we consider that it cannot be put in the paper since the survey would be closed.
eOr under your best criteria we would agree to any recommendation
https://docs.google.com/forms/d/e/1FAIpQLSfq7BMErQsWnraZsVmQZzCytWwwh_x965KZ2YoHaxWhkL2Tfw/viewform?usp=sf_link
The discussion needs to be extended, mainly regarding the Impact on changes in sleep schedules during the COVID-19 pandemic. More studies should be provided and discussed.
We have expanded the discussion section as well as the introduction section. In such a way that a more in-depth review of the subject is presented
Round 2
Reviewer 1 Report
Authors have conducted research on the eating habits and sleep quality during the COVID-19 pandemic in adult population of Ecuador. Authors have addressed all comments and suggestion given. This manuscript has been significantly improved. Well done!!!
This manuscript is now at a better quality to be published in the IJERPH. However, I found minor formatting issues that can be addressed to further improve this manuscript.
- Line 30, please add a space between people and the > sign.
- Table 5 heading should be on the same page as the table.
- Table 5: under the sub-section sleep disturbances, please format the number 69.2.
- Line 242, add the word "as" before the word "well"
Reviewer 2 Report
Dear authors, you have dealt with the requested amendments properly and correctly. I consider the manuscript suitable for publication in its present form.
I congratulate you.
Reviewer 4 Report
The authors have addressed my comments and the manuscript has been really improved. Now, it can be considered for publication in IJERPH.